# Category-orthogonal object features guide information processing in recurrent neural networks trained for object categorization

**Sushrut Thorat**[*]
sushrut.thorat94@gmail.com

**Giacomo Aldegheri**[*]
giacomo.aldegheri@gmail.com

**Tim C. Kietzmann**
tim.kietzmann@donders.ru.nl

Donders Institute for Brain, Cognition and Behaviour
6525 AJ Nijmegen, The Netherlands

## Abstract

Recurrent neural networks (RNNs) have been shown to perform better than feedforward architectures in visual object categorization tasks, especially in challenging conditions such as cluttered images. However, little is known about the exact computational role of recurrent information flow in these conditions. Here we test RNNs trained for object categorization on the hypothesis that recurrence iteratively aids object categorization via the communication of category-orthogonal *auxiliary* variables (the location, orientation, and scale of the object). Using diagnostic linear readouts, we find that: (a) information about auxiliary variables increases across time in all network layers, (b) this information is indeed present in the recurrent information flow, and (c) its manipulation significantly affects task performance. These observations confirm the hypothesis that category-orthogonal auxiliary variable information is conveyed through recurrent connectivity and is used to optimize category inference in cluttered environments.[†]

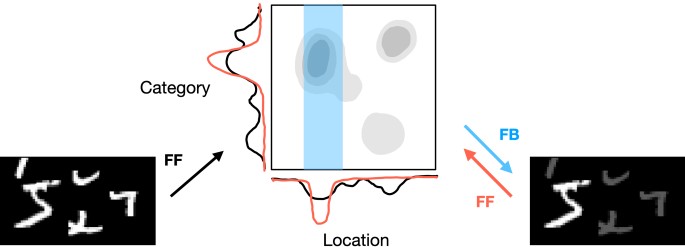

Figure 1: In cluttered images, the feedforward sweep (FF) of a recurrent neural network, trained for view-invariant object recognition, could learn to infer the location of the intact object (category-orthogonal, auxiliary variable) in addition to its category to filter out information from the irrelevant locations in the image (through feedback, FB), to improve the inference of the object's category.

---

[*]Equal contribution.
[†]Code and data: `https://github.com/KietzmannLab/svrhm21_RNN_explain`

3rd Workshop on Shared Visual Representations in Human and Machine Intelligence (SVRHM 2021) of the Neural Information Processing Systems (NeurIPS) conference, Virtual.

# 1    Introduction

While feedforward neural networks (FNNs) have demonstrated far-reaching success in the task of visual object categorization, recurrent neural networks (RNNs), inspired by the abundance and usefulness of recurrent connectivity in the primate visual system (Kar et al., 2019; Kietzmann et al., 2019), have been shown to outperform them in some settings (Spoerer et al., 2020; Kubilius et al., 2018). This advantage manifests particularly under challenging conditions such as partial object occlusion and clutter (Spoerer et al., 2017; Ernst et al., 2019). However, beyond the empirical finding that recurrence can help object categorization, it remains unclear what information is conveyed by recurrent connections and what its functional role is.

In contrast to FNNs, unit activations in RNNs are a function of both their input and their prior activations. This enables these networks to process input in time-varying, context-dependent, and conditional manner. The usefulness of these conditional computations has been observed, for example, in networks processing natural language, in which contextual words, such as negations, can 'steer' a network's trajectory in state space to process subsequent words differently (Maheswaranathan & Sussillo, 2020). In the primate visual cortex, recurrence is believed to underlie computations that can benefit from contextual signals, such as assigning local features to a figure or the background based on global shape consistency (Roelfsema et al., 2007; van Bergen & Kriegeskorte, 2020).

In object categorization, contextual modulations would likely take advantage of the structure of real-world data, similar to how recurrence can exploit the part-whole hierarchy of visual shapes in figure-ground segmentation or the branching structure of phrases in natural language. In particular, natural images are a function of both the categories of objects therein and other category-orthogonal *auxiliary* variables, such as the objects' location, orientation or scale. A potential role of recurrent connectivity could be to aid the selection of information that is most relevant for object categorization, by first extracting auxiliary variables about the object and subsequently to condition information processing on that information. That is, such a mechanism could iteratively focus on the location and features corresponding to the object in the image and filter out irrelevant noise such as clutter. That is, auxiliary, category-orthogonal information would not be discarded due to it being non-diagnostic for the category identity, but rather recurrent connectivity would use this information to guide and improve performance of the main task of object categorization (Fig. 1).

To test this hypothesis, we trained and tested multiple instances of an RNN on an object categorization task while presenting target objects in cluttered environments. We used diagnostic readouts across layers and time to characterise the presence of information related to auxiliary variables, and performed in-silico causal experiments to further elucidate their computational role in object categorization.

# 2    Methods

Primary details about the network architecture and the datasets are mentioned below. Please refer to the Appendix for exhaustive details.

## 2.1    Network architecture

The recurrent network architecture used for training and subsequent analyses consisted of two convolutional layers and one fully connected layer, as illustrated in Fig. 2A. The architecture contained both lateral and top-down recurrent connections. Top-down connections were sent from a given layer to all the previous layers, including the input. The RNN was unrolled for 4 timesteps. The activations of the fully-connected layers at the end of each timestep were concatenated and mapped to the classification output with a fully-connected layer.

## 2.2    Combining the feedforward and recurrent information flow

We operationalize the effect of recurrent connectivity in terms of gain modulation (multiplicative interaction). For this, we summed the incoming lateral and top-down recurrent activations, and multiplied them, element-wise, with the feedforward activations. We refer to the summed recurrent

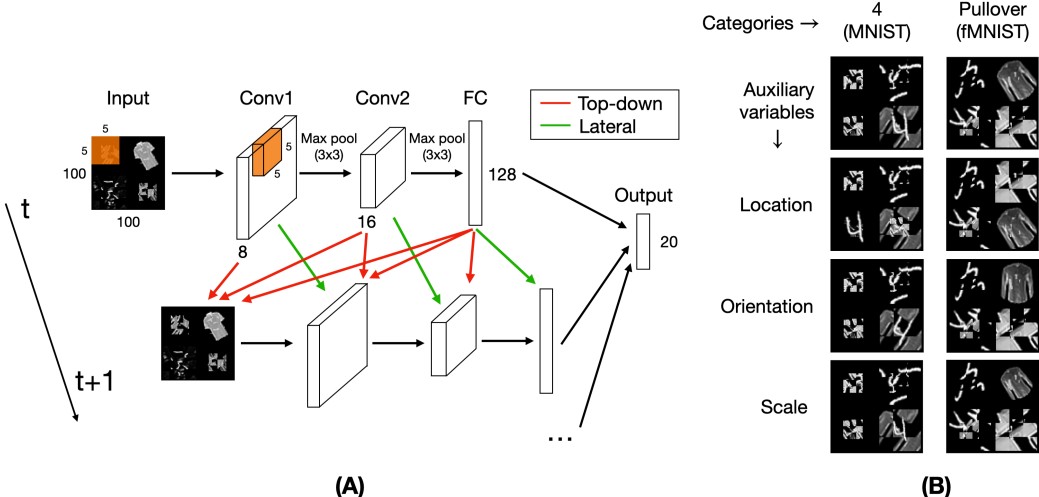

Figure 2: (A) Network architecture. Our RNN instances with multiplicatively interacting feedforward and recurrent information flows were unrolled for $4$ timesteps. (B) Two example images from the dataset with corresponding transformations along the auxiliary variables.

activations fed to a layer as the *recurrent information flow* to that layer. For a layer $l$ at timestep $t$ the activation was given by:

$$y_t^l = ReLU(C_{ff}(y_t^{l-1})) \odot \underbrace{(1 + C_{lat}(y_{t-1}^l) + \sum_{m>l} C_{td}^m(y_{t-1}^m))}_{\text{Recurrent information flow to layer } l \text{ at timestep } t} \tag{1}$$

where $C_{ff}$, $C_{lat}$ and $C_{td}$ correspond to the feedforward, lateral and top-down transformations corresponding to layer $l$ respectively. Activations at the Input layer were clipped between $0$ and $1$, to ensure the modulated input images always remained in the same range across timesteps.

## 2.3 Dataset and task

In order to study the effects of recurrent information flow, we trained our RNNs under challenging visual conditions. Specifically, we generated a dataset in which target objects were manipulated according to a number of category-orthogonal variables. Object categories were taken from the MNIST (LeCun et al., 1998) and Fashion-MNIST (Xiao et al., 2017) datasets - corresponding to 20 categories in total. The objects were randomly varied in location, orientation and scale. In addition, structured clutter, i.e. randomly sampled fragments of other objects in the dataset, was added to the images. Each of the auxiliary variables (horizontal and vertical locations, orientation and scale; Fig. 2B) had two possible values. In the results, the measures for the vertical and horizontal locations are averaged and summarized as one auxiliary variable (*location*).

The RNN was trained for 20-way classification. 5 instances of the RNN were trained from random initializations to assess the robustness of our findings (Mehrer et al., 2020).

## 3 Results

The networks' accuracy (averaged across 5 trained instances of the network) on a test set ($10,000$ images) was $81.2\%$ (chance performance for 20-way classification is $5\%$), implying that the network successfully learned to classify the images in the dataset. Next, we analyzed the networks' activations to assess whether auxiliary variables were extracted or suppressed, and how those variables affected information flow and network performance.

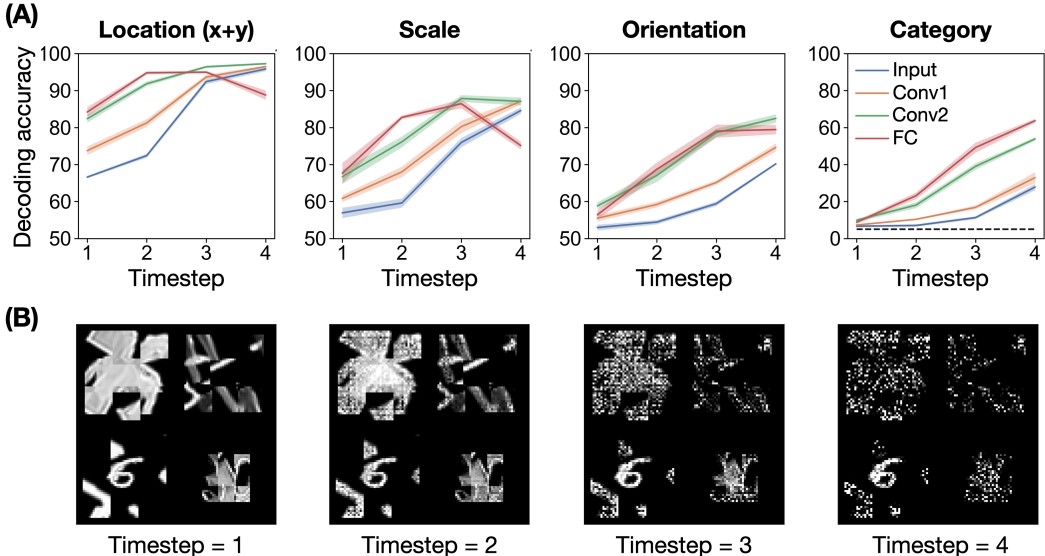

Figure 3: (A) Auxiliary variable information is observed in all the layers of the RNN at all the timesteps. $95\%$ confidence intervals of the average (across 5 RNN instances) accuracies are shown. (B) Clutter-reduction at the Input - an emergent phenomenon.

### 3.1 Category-orthogonal information is expressed incrementally over time in the RNN

To test for the presence of category-orthogonal information in the RNN activation patterns across layers and timesteps, we trained linear diagnostic readouts targeting auxiliary variables (i.e. predicting the location, scale, and orientation of the object), in addition to determining the presence of category information in the network activations. Results are shown in Fig. 3A.

First, our analyses revealed that category information increased with both layer depth (related to hierarchical processing) and time, indicating that the recurrent information flow carried information relevant to categorization at each timestep. Importantly, the values of all auxiliary variables could be decoded in all layers and at all timesteps, with increasing performance with increasing timesteps (averaged across layers, network instances, and auxiliary variables: $65.2\%, 73.0\%, 82.5\%, 84.9\%$ for timesteps 1-4). In summary, our analyses revealed that, instead of filtering out the category-orthogonal information, the RNNs extracted auxiliary variable information incrementally across time.

Importantly, auxiliary variable information (and also category information) could also be decoded increasingly well from the input image. This is due to the fact that the RNNs were set up to feed back information all the way down to the input, and as a result, the corresponding feedback effects can be easily visualised. As can be seen in Fig. 3B, incremental decoding of auxiliary variables was accompanied by clutter-reduction over time.

### 3.2 Recurrent information flow includes category-orthogonal information that guides subsequent network inference

To determine whether the auxiliary variable information was encoded in the recurrent information flow, we decoded all variables at each timestep from recurrent information only (starting from the second timepoint at which recurrence comes into effect). We found that the information about both auxiliary variables and category increased over time in the recurrent information flow to all layers, consistent with the information in the layers' activations reported earlier (see Fig. S1).

Does successful decoding of auxiliary variables from recurrent information flow imply functional impact, or is it a side-effect of layer activations with no causal role in the networks' categorization performance? To answer this question, we conducted a perturbation analysis in which we exchanged the feedback to a given layer and timepoint with feedback extracted from another, systematically perturbed, image from the dataset (Fig. 4A), i.e. feedback signalling the wrong value of the auxiliary variable. As any manipulation may alter network performance, we furthermore included a control condition in which we exchanged the original feedback signal with a randomly rotated version of the

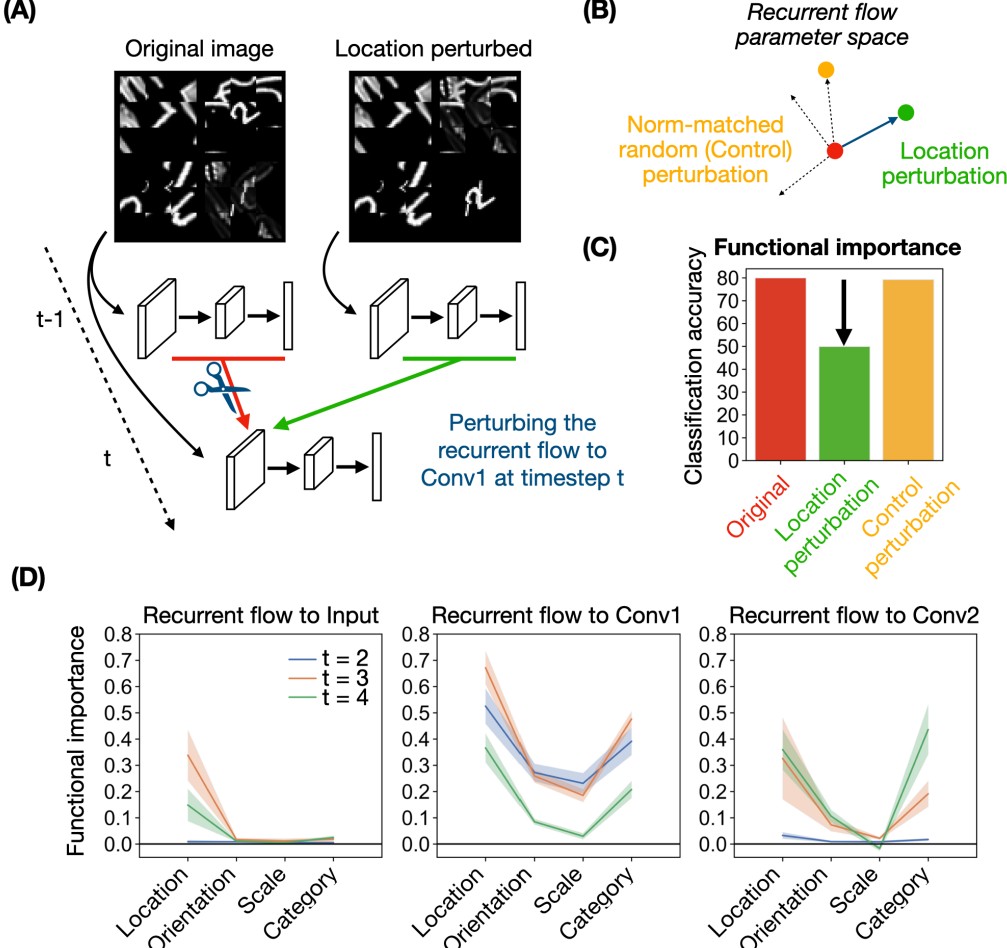

Figure 4: (A) An example perturbation: perturbing the object's location and injecting the resulting recurrent information flow into the network, at Conv1, processing the original image at timestep $t$. (B) Illustration of a control perturbation matched in magnitude to the systematic perturbation corresponding to a specific variable (location). (C) Functional importance = (Performance of the control perturbation - Performance of the systematic perturbation) / Original performance. (D) Functional importance of perturbing the auxiliary variables at different layers and timesteps. Location information is functionally important in the recurrent information flow to the Input at later stages of processing. In addition to location, orientation, scale, and category are functionally important in the recurrent information flow to the convolutional layers. Average data across 5 RNN instances shown together with $95\%$ confidence intervals.

perturbation vector (Fig. 4B corresponding to the systematic perturbation, see Appendix for details). This control perturbation signals different, but not entirely misleading, information about the auxiliary variables (as it does not consistently align with changes in any of the auxiliary variables).

The functional importance of a variable in the recurrent information flow was quantified as follows: first, we contrasted the networks' task performance following two types of recurrent information flow perturbation: systematic and control. That is, we computed how much more a systematic perturbation affected task accuracy compared to a control perturbation. This difference was then normalised based on the original network performance. As a result, functional importance was computed as $(Accuracy_{control} - Accuracy_{systematic})/Accuracy_{original}$ (Fig. 4C).

This analysis demonstrated that location information in the recurrent information flow to the Input was functionally important at timesteps 3 and 4 (Fig. 4D), which corresponded to the major part of the clutter-reduction observed in Fig. 3B. In addition to location, information about other auxiliary object features - orientation, scale, and category - was functionally relevant in the recurrent information

flow to the convolutional layers. Neither the auxiliary variables nor category information in the lateral information flow to FC were functionally important (data not shown). In line with this observation, ablating that lateral connection from the trained RNN led to no performance reduction. In summary, auxiliary variable information in the recurrent information flows seems to causally guide the information processing in the network to support object categorization.

# 4 Discussion

We investigated the role of recurrent information flow in RNNs trained for object categorization in cluttered environments. We hypothesized that, to improve categorization performance, category-orthogonal variables are extracted, rather than filtered out, and subsequently used by the RNN to constrain later information processing. Consistent with this hypothesis, we found that (i) information about all auxiliary variables was present at all network layers, (ii) this information became more prominent across time, and (iii) perturbing this information in the recurrent information flow significantly reduced network performance.

The task of object categorization has traditionally been cast in terms of extracting representations invariant to all category-orthogonal variables. However, extracting auxiliary variables from visual input might be important for natural organisms, who must also be able to keep track of the location and other properties of objects in order to survive (for instance, where a predator is and whether it is asleep or awake). This can explain the finding that primate inferior temporal cortex does contain such auxiliary variable information (Hong et al., 2016). In that study, surprisingly, information about the auxiliary variables was also found in a feedforward neural network trained exclusively for object categorization. That finding, and our current results, echo several proposals that suggest that optimally separating object categories might in fact require explicitly extracting auxiliary variables that characterize the variation of the objects in their images (DiCarlo & Cox, 2007; Patel et al., 2016).

As proposed in the introduction, once auxiliary variable information has been extracted, it can be used to improve categorization performance by conditioning category inference on the values of the auxiliary variables (Fig. 1). RNNs are particularly suited for this, since their architecture provides separate channels for inference (feedforward information flow) and conditioning (recurrent information flow) - an inductive bias that matches the proposed interaction between auxiliary variables and category inference. This inductive bias might be the reason recurrent architectures outperform parameter-matched feedforward architectures, particularly when the category inference is ambiguous - such as in the presence of clutter - where iterative conditioning from the auxiliary variables might be beneficial (Spoerer et al., 2017; Ernst et al., 2019). Additionally, *top-down* recurrent connections might be advantageous by allowing the networks to condition the inference at different hierarchical levels. This might be particularly important in the case of cluttered images since the lower spatial resolution of later layers might not allow the disentanglement of the target object from the clutter, and information selection might thus be more effective at earlier layers.

Relatedly, residual blocks, a popular architectural pattern in feedforward models, might provide an inductive bias similar to recurrence (Liao & Poggio, 2016), enabling an equivalent form of iterative processing, which could explain their increased efficacy in categorization (compared to vanilla feedforward models), especially for ambiguous images (Jastrzębski et al., 2017).

Another relevant inductive bias in our networks is the way in which feedforward and recurrent information interact. We focused our main analyses on an RNN architecture with multiplicative feedback interaction. In an RNN with additive interactions, we observed similar results in terms of the decodability of auxiliary variables across time and layers (see Appendix). The usefulness of the extraction of auxiliary variables therefore seems to be independent of the mode of interaction. However, with the additive interactions, we could not observe similarly strong clutter reduction in the input over time. The solution found by the multiplicative interactions for combining recurrent and feedforward information was therefore different, aligning more closely with our intuitive notion of information selection. An exact characterization of this distinction between the two types of interactions is beyond the scope of this study. However, it is useful to note that multiplicative interactions have been proposed to be a useful inductive bias for conditional computations (Jayakumar et al., 2020), so characterizing their unique role in how information can be conditioned with auxiliary variables for object categorization, is a promising avenue for future research.

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

## A  Appendix

### A.1  Dataset generation

Each image in our dataset was $100 \times 100$ pixels in size. Each image contained one intact target object (originally $28 \times 28$ px), rotated and scaled. 7 other *scrambled* objects comprised the clutter, each of which were also rotated and scaled, then divided into 9 square blocks, and permuted randomly. The image was divided into four quadrants, corresponding to the two horizontal and two vertical locations. Each quadrant contained two objects - one overlaid on top of the other. One of the quadrants contained the intact object which was always overlaid on the scrambled object, to ensure it wouldn't

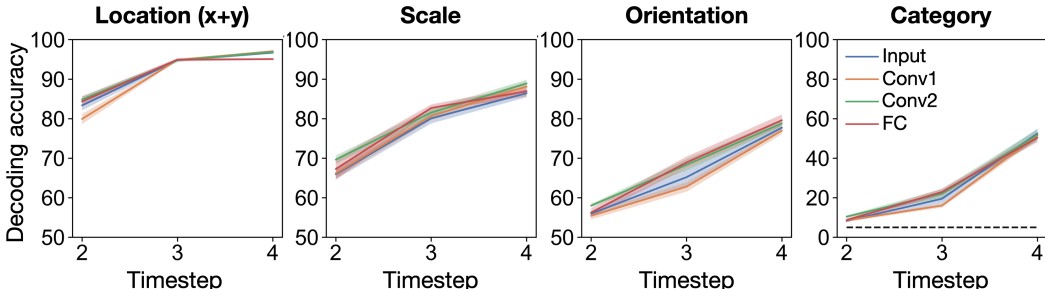

Figure S1: Auxiliary variable and category information is observed in all recurrent information flow, targeting all layers at all timesteps. To assess if the auxiliary variable and category information observed influences the subsequent feedforward sweep, the perturbation analysis shown in Fig. 4, was performed. 95% confidence intervals of the average (across 5 RNN instances) accuracies are shown.

be completely occluded while still being challenging to parse from the background. Both target and scrambled objects could be drawn with equal probability from either the MNIST or Fashion-MNIST dataset.

The horizontal and vertical location of the target object was chosen randomly for each image, meaning that the object would be located at the center of one of the four quadrants, jittered both horizontally and vertically between $-2.5$ and $2.5$ pixels (uniform distribution). The orientation of the objects (both target and scrambled) was randomly chosen to be either 30 degrees clockwise or counterclockwise, with a uniformly distributed jitter of $\pm 5$ pixels. Scaling was either $0.9$ or $1.5$ times the original object size, with a jitter of $\pm 0.1$.

The split between train, validation and test sets was done by randomly drawing images only from the corresponding splits of the original datasets. The datasets provided the $110,000$ possible source images for the training set (MNIST and Fashion-MNIST combined), $10,000$ for the validation set and $20,000$ for the test set. Training images were generated on the fly for each training batch. The test set contained $10,000$ images.

## A.2 Architecture details

The RNN contained two convolutional layers, both with $5 \times 5$ kernels, stride of 1, no padding, and with 8 and 16 channels respectively, and one fully-connected layer with 128 units. $3 \times 3$ max pooling was applied after each convolutional layer. The lateral recurrent connections from each convolutional layer to itself were convolutional layers with the same number of input and output channels (8 and 16), kernel size $5 \times 5$, stride 1 and padding 2. The lateral connection for the fully connected layer was a fully connected layer with 128 units. The top-down connections from convolutional layers to the input and other convolutional layers were transposed convolutional layers, upsampling to the size of the target feature map: Conv1 to Input had kernel size $7 \times 7$, stride 3, and no padding; Conv2 to Input had kernel size $20 \times 20$ stride 10 and no padding; Conv2 to Conv1 had kernel size $16 \times 16$, stride 10 and no padding. Top-down connections from the fully-connected layer were fully-connected layers whose outputs were restructured to match the convolutional layers.

## A.3 Training the RNN

The network was trained for 20-way classification using a cross-entropy loss between the network's output and the correct object category. The training images were generated on the fly. We used the Adam optimizer for training, with a batch size of 32, and learning rate of $10^{-4}$ (after manual tuning to get the best validation accuracy). The network was trained for $300,000$ iterations. Before training, the weights were initialized using Xavier initialization and the biases were initialized as zeros.

## A.4 Decoding approach

To measure the amount of explicit information about a given variable present in the activations, at each layer and at each timestep, we trained and tested a linear classifier (a support vector machine, SVM) to classify the correct value of that variable. We used the Python scikit-learn function

`svm.LinearSVC()`, with default parameters, which is able to handle binary as well as multi-class classification, which was needed to decode category. Each classifier was trained with $800$ images and tested on $200$ images (all randomly drawn from the test set). This procedure was repeated $10$ times (accuracies averaged), for each of the $5$ RNN instances.

### A.5 Perturbation analysis

To generate the *perturbed* feedback to the network at a given timestep $t$ and layer $l$, we ran two copies of the network. One was fed the original input image, and the other the same image, altered in one particular variable (auxiliary or category). For example, vertical location could be changed from top to bottom (Fig. 4A). From this network we extracted the perturbed recurrent information flow to $l$ at $t$, and then fed this recurrent information flow, at timestep $t$, to layer $l$ of the network which had received the original image, thus combining the activations for the original image with the recurrent information flow resulting from the perturbed image. We then compared this network's systematically perturbed classification accuracy to the network's accuracy for the same image without any perturbations.

To control for the unique role of the variables in determining network performance, we also measured the accuracy of a network that received the same perturbed recurrent information flow, but with the elements of the recurrent information flow vector randomly permuted (control recurrent information flow). This corresponds to a recurrent information flow that differs from the original one by the same amount (vector length) as the relevant perturbed recurrent information flow, but along a direction that, on average, does not align with any of the relevant variables (Fig. 4B).

The functional importance score ($\frac{control - systematic\ perturbation}{original}$, Fig. 4C) for each variable manipulation was computed as the average across $1000$ images (randomly drawn from the test set). This procedure was repeated $5$ times (scores averaged), for each of the $5$ RNN instances.

In the case of category, two types of perturbations were analyzed. Either the category of the intact object was changed within its dataset (within-domain perturbation, e.g. 3 changed to a 4, or 'trouser' changed to 'dress') or the category was changed across datasets (between-domain perturbation, e.g. 3 changed to a 'trouser'). In the results shown in the main text and in Fig. S2C, the averaged functional importance scores of these two types of category perturbations were shown. For both the additive and multiplicative interactions, the functional importance of the domain information of the category (indexed by the between-domain perturbation) was equivalent or higher than the within-domain information of the object category (which was functionally important at all the layers and timesteps where domain information was functionally relevant).

### A.6 Results for additive recurrent interactions

The additive interactions between the feedforward and recurrent information flow were implemented as follows: for a layer $l$ at timestep $t$ the activation was given by:

$$y_t^l = ReLU(C_{ff}(y_t^{l-1}) + \underbrace{C_{lat}(y_{t-1}^l) + \sum_{m>l} C_{td}^m(y_{t-1}^m)}_{\text{Recurrent information flow to layer } l \text{ at timestep } t}) \tag{2}$$

where $C_{ff}$, $C_{lat}$ and $C_{td}$ correspond to the feedforward, lateral and top-down transformations corresponding to layer $l$ respectively. All the other details of the network architecture and training were identical to the network with multiplicative interactions.

The profile of results for the network with additive recurrent interactions had both similarities and differences to the multiplicative ones. The decoding of variables at all layers and timesteps showed a similar profile (Fig. S2A), but in visualizing the modulation at the input image level, no clutter reduction comparable to that in the multiplicative network was visible (Fig. S2B). Consistent with this observation, no variable was found to be functionally important in the recurrent information flow to the input, while other layers showed a pattern of variable importance similar to that of the network with multiplicative recurrent interactions (Fig. S2C).

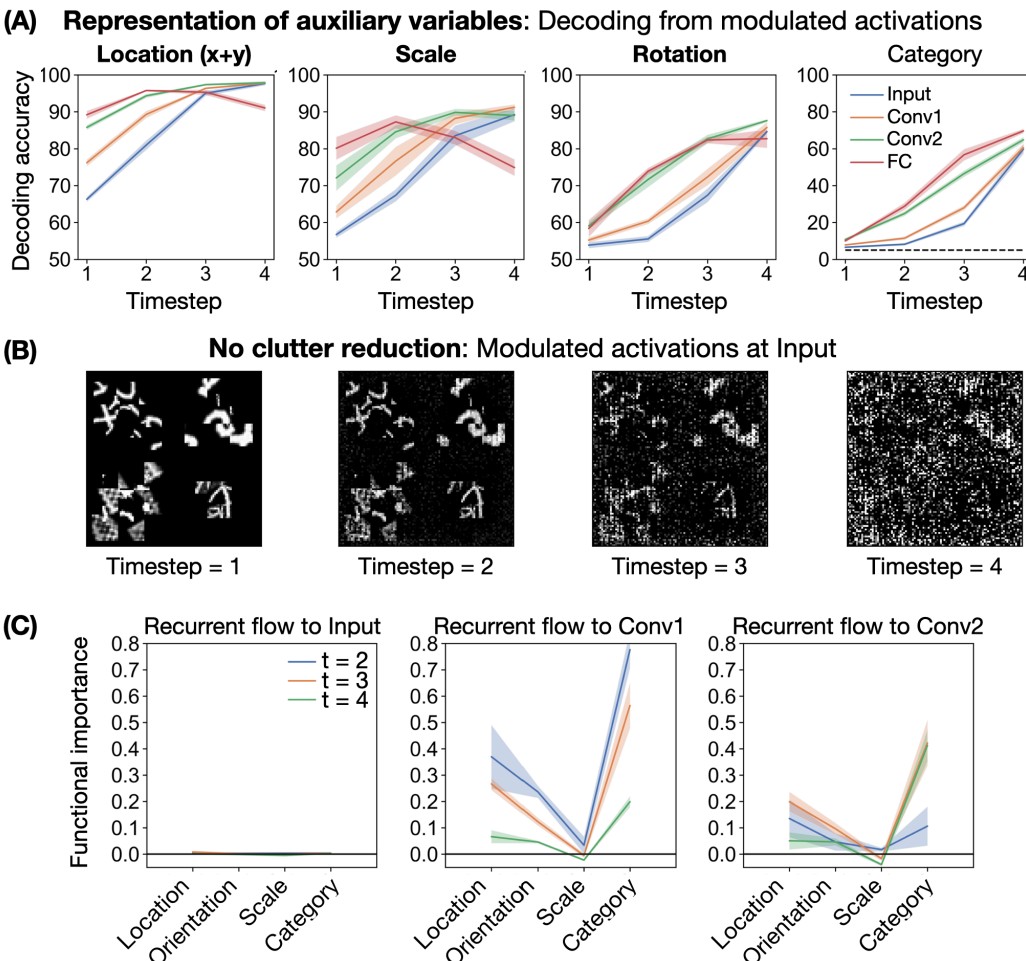

Figure S2: (A) Auxiliary variable information in all layers of the network with additive recurrent interactions, at all timesteps. 95% confidence intervals of the average (across 5 RNN instances) accuracies are shown. (B) No clutter reduction is observed at the Input. (C) Functional importance of the variables at each layer and timestep. Consistent with the lack of clutter reduction, no variable was found to be functionally important in the recurrent information flow to the Input.

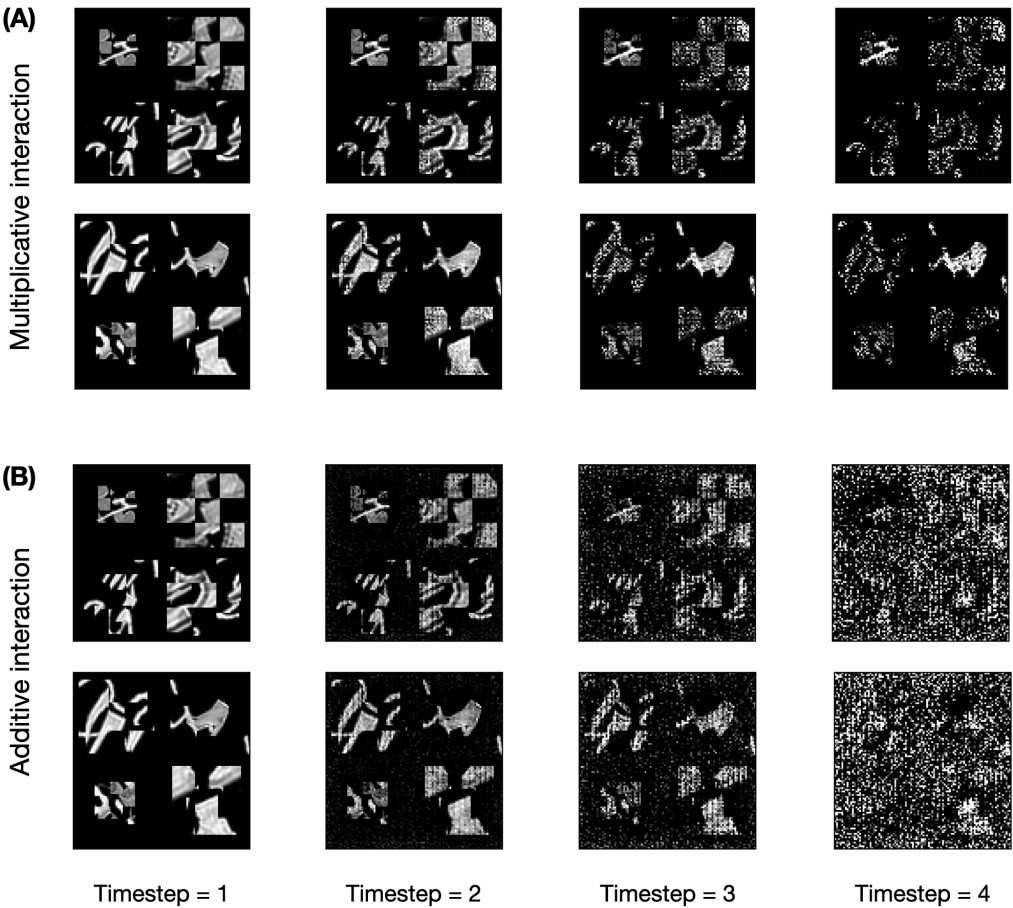

Figure S3: (A) More examples of input-level modulation for the RNN with multiplicative interactions. (B) Examples for the RNN with additive interactions. Both the RNNs with multiplicative or additive interactions classified both these images correctly, but with different strategies.

