# OpenReview forum: "Category-orthogonal object features guide information processing in recurrent neural networks trained for object categorization"
_NeurIPS.cc/2021/Workshop/SVRHM — SVRHM 2021 Poster_

### Official Review · Reviewer_oXeB · 2021-10-20
**Review for "Category-orthogonal object features guide information processing in recurrent neural networks trained for object categorization"**

**Rating:** 8
**Confidence:** 3

**Review:**

## Summary:

In the present paper, the authors test the hypothesis that the role of recurrent connectivity in object-categorization is to convey category-orthogonal information such as location or orientation which aids in recognizing a given object, particularly in challenging conditions such as cluttered environments. The authors aim to test this hypothesis by training a simple recurrent convolutional neural network on an object-categorization task with cluttered images and evaluating the performance as well as the presence of category-orthogonal information in the network across layers and time steps. They show that category can be read out from the recurrent network as well as category-orthogonal information across layers and increasingly across time steps. Further they show that this category-orthogonal infromation is conveyed by feedback and that feedback information flow is functionally relevant for the object-categorization task.

## Pros:

- the hypothesis and results are concisely communicated and the logic of the paper is easy to follow
- the perturbation analyses nicely demonstrate that the auxiliary-variable information is not only a side-effect of the categorization training of the network but that it is directly related to task perfromance and clutter reduction
- visualization of the methods and results is clear and supports readability of the paper

## Cons:

- to me it was not entirely clear where the formalization of the recurrent flow is coming from. Is that idea based on ideas from neuroscience ? Or is that a theory that the authors came up with by themselves ? It would be nice to read more about why the authors think that this is a good quantification of recurrent information flow

## Overall summary:

In sum, I really enjoyed reading the paper. It deals with an intriguing question, uses a well defined method and clear tests to investigate the question and the results clearly align with the conclusions by the authors. I definitely recommend accepting the paper.

---

> ### Author Response · Authors · 2021-11-15
> **Response**
>
> Thank you for your comments, and for your kind words.
> In response to your concern:
>
> > to me it was not entirely clear where the formalization of the recurrent flow is coming from. Is that idea based on ideas from neuroscience ? Or is that a theory that the authors came up with by themselves ? It would be nice to read more about why the authors think that this is a good quantification of recurrent information flow
>
> We have now made more explicit in the paper the link with neuroscience: part of the inspiration came from the abundance of recurrent connections in the visual cortex, the fact that these play a role in modulating feedforward processing, and the explicit extraction of auxiliary variables in the primate ventral stream (e.g. https://www.nature.com/articles/nn.4247).
>
> But it was mostly a more theoretical question: given the empirical finding that RNNs can outperform feedforward networks in object classification, what operations are they implementing? Since natural images are a function of both category and category-orthogonal (auxiliary) variables, and inference about category can be conditioned on the value of auxiliary variables, we hypothesized the role of recurrence might be to implement this conditioning operation. We tried to make this line of reasoning, too, more clear in the Discussion.

---

### Official Review · Reviewer_hptj · 2021-10-20
**Insights into the computational role of recurrence and feedback in vision**

**Rating:** 8
**Confidence:** 4

**Review:**

## Summary
The authors aim to understand the computational role recurrence and feedback in visual object categorization. They note that while multiple studies have demonstrated the benefits of recurrence in cases such as cluttered environments, we know little about the mechanisms involved. They then propose the hypothesis that recurrence amplifies information about category-auxiliary variables (e.g. object location) and uses it to modulate activity across the model. In this way, a model can selectively attend to regions in the image that contain the object of interest. Finally, they confirm this hypothesis through linear readouts of model activations to predict category-auxiliary variables.

I really enjoyed this paper. It had a clear well-motivated hypothesis and used straight-forward methods to test it. As a result, the conclusions are convincing and valuable for understanding the computational role of recurrent processing in vision.

## Positives
- Due to the simplicity of the architecture, recurrent layers, dataset, and linear decoders, it's hard to poke holes in any of the analyses or conclusions. The results also show a very significant and consistent effect of time on the decodability of all category-auxiliary variables considered.
- The dataset was nicely constructed and seems to be a great benchmark for classification tasks in cluttered environments.
- I really like the idea of adding recurrence back to the input as a way of visualizing its effect. It makes it clear that the category-auxiliary variables are likely being used to amplify the region of interest and remove other clutter.

## Negatives
- My biggest concern is the lack of any purely feed-forward benchmark. The argument is that recurrence is beneficial for extracting and using category-auxiliary varialbes (which I believe), but can a feedforward model do the same? Granted, at timestep t=1 the model has only performed feed-forward computations, but it was still trained with recurrence and might rely on future timesteps to better extract category-auxiliary varialbes. The representations in a model that is only ever trained with feed-forward connections might be very differently. Furthermore, if the depth of a feedforward model is made equal to that of the unrolled RNN, does recurrence still provide an advantage in terms of task performance and decodability of category-auxiliary variables?
- Category-auxiliary variables should be defined earlier on. I was initially unsure as to the meaning, and I think that naming them explicitly in the abstract (location, scale, orientation) would resolve this issue.
- The main text doesn't mention what transformations are used for lateral or feedback recurrence ($C_{lat}$ and $C_{td}^m$). The appendix makes it clear that these are convolutional, fully-connected, or transposed convolutional layers, but I think it would be best to quickly mention this in the main text for clarity.
- The reasons for emphasizing recurrence in classification tasks in particular wasn't obvious to me. In fact, the proposed category-auxiliary variables and their use in modulating network activity seems relevant for spacial attention more generally. In other words, instead of arguing that recurrent processing enables the extraction and use of category-orthogonal variables in categorization, a more general argument could have been made for the role of recurrent processing in tasks that require spatial attention.

---

> ### Author Response · Authors · 2021-11-15
> **Response**
>
> Thank you for your feedback and kind words on the paper. We address your concerns below:
>
> > My biggest concern is the lack of any purely feed-forward benchmark. The argument is that recurrence is beneficial for extracting and using category-auxiliary varialbes (which I believe), but can a feedforward model do the same? Granted, at timestep t=1 the model has only performed feed-forward computations, but it was still trained with recurrence and might rely on future timesteps to better extract category-auxiliary varialbes. The representations in a model that is only ever trained with feed-forward connections might be very differently. Furthermore, if the depth of a feedforward model is made equal to that of the unrolled RNN, does recurrence still provide an advantage in terms of task performance and decodability of category-auxiliary variables?
>
> Our aim was not to claim a general performance advantage of RNNs vs. feedforward networks. Rather, we started from the empirical finding that this advantage can occur in a particular task (object categorization, especially in cluttered images), and tried to explain what operations might be at work in RNNs performing that task.
> We also do not believe the principle of explicitly extracting auxiliary variables is unique to RNNs (and we cite work showing that it also applies to feedforward nets, e.g. https://arxiv.org/abs/1504.00641, https://www.sciencedirect.com/science/article/pii/S1364661307001593, https://www.nature.com/articles/nn.4247) but rather that recurrence constitutes an inductive bias for _using_ that information to condition inferences about object category. We have now made this distinction more explicit in the Discussion.
> Also in the Discussion, we have added that it is possible for feedforward neural networks to implement similar forms of conditional, iterative processing using residual blocks, which have been found to approximate recurrence (https://arxiv.org/abs/1710.04773).
>
> > Category-auxiliary variables should be defined earlier on. I was initially unsure as to the meaning, and I think that naming them explicitly in the abstract (location, scale, orientation) would resolve this issue.
>
> We listed them explicitly in the abstract, indeed it's better to clarify that early in the text.
>
> > The main text doesn't mention what transformations are used for lateral or feedback recurrence ($C_{lat}$ and $C_{td}^m$). The appendix makes it clear that these are convolutional, fully-connected, or transposed convolutional layers, but I think it would be best to quickly mention this in the main text for clarity.
>
> We did not include this (and many other details about the methods) in the main text due to space limitations. We have added a pointer to the Appendix at the beginning of the Methods section.
>
> > The reasons for emphasizing recurrence in classification tasks in particular wasn't obvious to me. In fact, the proposed category-auxiliary variables and their use in modulating network activity seems relevant for spacial attention more generally. In other words, instead of arguing that recurrent processing enables the extraction and use of category-orthogonal variables in categorization, a more general argument could have been made for the role of recurrent processing in tasks that require spatial attention.
>
> As mentioned in the comment above, one of the starting points of our work was the empirical finding that RNNs can outperform feedforward networks in classification, and we wanted to clarify that finding. Classification is a particularly interesting task because in principle, it does not _require_ to extract auxiliary variables, contrary to e.g. image segmentation. Other tasks do not have such a sharp distinction between a variable of interest (category) and others (auxiliary variables) that may be discounted, so it isn't clear how conditioning on auxiliary variables could apply there.
> However, how our results can speak to other tasks is definitely an interesting question for follow-up work!

---

### Official Review · Reviewer_H1aF · 2021-11-01
**An interesting study on context-dependent processing in an RNN network; however, the relevance for biological and machine learning systems is not made clear.**

**Rating:** 6
**Confidence:** 4

**Review:**

The submission studies how category-orthogonal information guides object recognition in a small recurrent network (RNN). The authors hypothesize that category-orthogonal information (e.g., location, size) is not reduced during processing, but is instead conditioned on in order to process category-relevant information (e.g., shape). Two experiments (decodability and an ablation study) confirm this hypothesis.


Strengths:
- The question is meaningful and carefully investigated with a correlational study and a causal manipulation.

Weaknesses:
- The broader impact of the study is not made clear because the small RNN is not demonstrated to be representative of biological or modern machine learning systems.
  - It is mentioned in the paper that biological systems use recurrent connections, but the particular architectural studied in the paper is not motivated further; the system & problem setting is also far from object categorization systems used in current practice (ie. deep neural networks, even though MNIST and its variant is claimed to be "visually challenging").
- The experiments are missing some important explanations that are required to interpret the results:
  - I like the approach of using "linear readouts" to detect for "decodability" (thus availability) of the category and auxiliary information. However, the process for training the readout gadgets was insufficiently described: I am assuming they are linear classifiers but I don't believe this was described in the text. Moreover, how are train/test splits constructed for these gadgets? How is the target value produced?
  - The causal experiment is interesting and important to establish the functional importance of the auxiliary information, but the manipulations and baseline were not clear from the text, and so the experiment was less impactful than it could have been.

Minor:
- Some terminology is not sufficiently defined / explained:
  - "lateral connections"
  - "gain modulation"

---

> ### Author Response · Authors · 2021-11-15
> **Response**
>
> Thank you for your feedback. We address your concerns below:
>
> > The broader impact of the study is not made clear because the small RNN is not demonstrated to be representative of biological or modern machine learning systems.
>
> The inspiration for the study was partly coming from neuroscience, specifically the importance of feedback connections in modulating feedforward processing in the visual cortex, and the explicit representation of category-orthogonal variables in the primate ventral stream (e.g. see https://www.nature.com/articles/nn.4247, which we also cite in the paper).
> However, our aim was not to model any specific neural or behavioral finding, but rather to propose a general computational principle for object categorization, that might be at work in both biological and artificial neural networks.
> The assumption behind our task and dataset is quite general: that images are the result of both category and category-orthogonal variables. We used a highly simplified setting to be able to investigate the network's behavior in detail, but this assumption still holds in natural images. Of course, this does not mean that these results will trivially generalize to more naturalistic datasets, but they are meant as a starting point for follow-up work.
> We have now made more explicit in the Discussion both the neuroscientific inspiration, and how our findings can speak to it, and the links with modern deep learning architectures, such as ResNets, that suggest similar principles of conditional processing might be at work in bigger models processing natural images.
>
> > The experiments are missing some important explanations that are required to interpret the results
>
> All of the detailed information about both the decoding and the perturbation analysis can be found in the Appendix. Due to the 5-page limit, it was simply not possible to include all of this information in the main text. We have added at the beginning of the Methods a sentence pointing to the Appendix for exhaustive details.
>
> > Some terminology is not sufficiently defined / explained:
> "lateral connections"
> "gain modulation"
>
> We did not explicitly define lateral recurrent connections since we thought it was clear from the context that the term refers to recurrent connections from a layer to itself. Gain modulation is defined at the beginning of Section 2.2, as multiplicative interaction or element-wise multiplication.

---

### Decision · Program_Chairs · 2021-11-02

Accept (Poster)